# Integration of EHR and ECG Data for Predicting Paroxysmal Atrial Fibrillation in Stroke Patients

**DOI:** 10.3390/bioengineering12090961

**Published:** 2025-09-07

**Authors:** Alireza Vafaei Sadr, Manvita Mareboina, Diana Orabueze, Nandini Sarkar, Seyyed Sina Hejazian, Ajith Vemuri, Ravi Shah, Ankit Maheshwari, Ramin Zand, Vida Abedi

**Affiliations:** 1Department of Public Health Sciences, College of Medicine, Pennsylvania State University, Hershey, PA 17033, USA; asadr@pennstatehealth.psu.edu; 2Institute for Personalized Medicine, Department of Biochemistry and Molecular Biology, The Pennsylvania State University College of Medicine, Hershey, PA 17033, USA; mmareboina@pennstatehealth.psu.edu; 3Penn State Hershey Medical Center, Penn State College of Medicine, Hershey, PA 17033, USA; dorabueze@pennstatehealth.psu.edu (D.O.); nsarkar@pennstatehealth.psu.edu (N.S.); 4Department of Neurology, College of Medicine, The Pennsylvania State University, Hershey, PA 17033, USA; sina.hej95@gmail.com (S.S.H.); avemuri@pennstatehealth.psu.edu (A.V.); 5Division of Cardiology, Heart and Vascular Institute, Penn State Hershey Medical Center, Hershey, PA 17033, USA; rshah6@pennstatehealth.psu.edu (R.S.); amaheshwari@pennstatehealth.psu.edu (A.M.)

**Keywords:** paroxysmal atrial fibrillation, electronic health records, electrocardiogram, multimodal data integration, deep learning, feature importance analysis

## Abstract

Predicting paroxysmal atrial fibrillation (PAF) is challenging due to its transient nature. Existing methods often rely solely on electrocardiogram (ECG) waveforms or Electronic Health Record (EHR)-based clinical risk factors. We hypothesized that explicitly balancing the contributions of these heterogeneous data sources could improve prediction accuracy. We developed a Transformer-based deep learning model that integrates 12-lead ECG signals and 47 structured EHR variables from 189 patients with cryptogenic stroke, including 49 with PAF. By systematically varying the relative contributions of ECG and EHR data, we identified an optimal ratio for prediction. Best performance (accuracy: 0.70, sensitivity: 0.72, specificity: 0.87, Area Under Curve - Receiver Operating Characteristics (AUROC): 0.65, Area Under the Precision-Recall Curve (AUPRC): 0.43) was achieved using a 5-fold cross-validation when EHR data contributed one-third and ECG data two-thirds of the model’s input. This multimodal approach outperformed unimodal models, improving accuracy by 35% over EHR-only and 5% over ECG-only methods. Our results support the value of combining ECG and structured EHR information to improve accuracy and sensitivity in this pilot cohort, motivating validation in larger studies.

## 1. Introduction

Paroxysmal atrial fibrillation (PAF), a major stroke risk factor, characterized by transient arrhythmic episodes, poses significant diagnostic challenges due to its sporadic nature, with up to 40% of cases remaining asymptomatic [1,2]. Traditional detection methods, such as Holter monitoring, often fail to capture these fleeting events, leading to delayed diagnosis and increased risk of recurrent stroke and systemic embolism [3,4]. Recent advances in machine learning have enabled novel approaches to PAF prediction, with convolutional neural networks (CNNs) achieving higher AUROCs than conventional ML methods using electrograms or 12-lead electrocardiograms (ECGs) alone [5,6,7,8,9]. However, traditional (unimodal) models face limitations: ECG-based approaches primarily detect electrophysiological anomalies, while electronic health record (EHR)-driven models rely on systemic risk factors such as hypertension and diabetes, which lack key temporal resolution [7,10].

Integrating ECG and EHR data could synergistically enhance predictive accuracy by contextualizing transient ECG abnormalities within longitudinal clinical profiles, a hypothesis supported by studies showing improvement in diagnosis prediction when combining these data sources [11]. Multimodal deep learning frameworks, particularly those employing attention mechanisms, have demonstrated promise in cardiac applications; Transformer architectures excel at modeling long-range dependencies in sequential data, while CNNs capture localized ECG features such as P-wave morphology and QRS complex variations [12,13]. For instance, Tzou et al. [13] achieved high sensitivity for PAF prediction by analyzing P-wave dynamics and skin sympathetic nerve activity using a wavelet–CNN hybrid, while Tang et al. [7] reported strong performance for PAF reoccurrence by fusing intracardiac electrograms with clinical variables. However, no study has systematically determined the optimal integration of raw ECG waveforms and structured EHR data for predicting PAF, representing a significant gap given the episodic and elusive nature of the condition [2,10].

To address this gap, our pilot study deliberately focuses on dissecting the relative contributions of ECG and EHR data. The primary objective was to develop a multimodal deep learning model for predicting PAF. We hypothesized that a multimodal deep learning model would outperform unimodal models by optimally balancing ECG and EHR contributions. We introduce a Transformer-based architecture that integrates denoised 12-lead ECG signals with 47 EHR parameters, including cardiac monitoring duration and hemoglobin levels. This approach quantifies the optimal ratio between ECG and EHR. Class imbalance is addressed through stochastic data augmentation techniques, time warping, amplitude scaling, and Gaussian noise injection [14]. We further discuss implications for early intervention strategies, address limitations in sensitivity, and propose future directions, including latent EHR feature exploration and prospective and external validation. This work advances personalized PAF management by bridging electrophysiological and systemic risk assessment through explainable multimodal learning [12,15]. This work is intended to support outpatient rhythm monitoring triage and not to replace clinician judgment.

## 2. Materials and Methods

### 2.1. Study Design and Data Collection

This study utilized a dataset comprising 189 cryptogenic stroke patients, collected by medical students at an academic hospital, Penn State College of Medicine, from January 2017 to May 2023 under an exempt Institutional Review Board protocol. The dataset included both ECG waveforms and EHR data. All data were validated by a cardiologist and a stroke neurologist to ensure diagnostic accuracy and clinical relevance. Reporting followed the TRIPOD-AI guideline; a completed checklist is provided in the Supplement (Table A1).

### 2.2. Inclusion/Exclusion

Inclusion criteria were adults ≥ 18 with cryptogenic ischemic stroke; participants were excluded if they had persistent/permanent AF before index ECG, non-12-lead ECG, or missing EHR core variables.

### 2.3. Data Preprocessing

ECG waveforms were extracted from XML files. Each 12-lead ECG signal was normalized to a range of 0–1 and preprocessed using wavelet denoising and bandpass filtering (0.5–40 Hz). Multiple representations of ECG data were explored, including raw signals, denoised signals, and denoised–filtered signals. Data augmentation techniques were applied to the ECG signals to address class imbalance, including time warping, amplitude scaling, baseline wander addition, Gaussian noise injection, random permutation, and random shifting. The augmentation probability (P_aug_) and the number of augmented samples were optimized as hyperparameters.

We selected 47 clinically relevant EHR parameters based on known PAF risk factors and predictors. We used information available at the index stroke encounter. Predictors included demographics (age, sex, race, ethnicity); BMI (body mass index); comorbidities/new diagnoses at index (hypertension, type 2 diabetes, hyperlipidemia, hypercoagulability, chronic kidney disease, liver disease, active cancer, prior ischemic/hemorrhagic stroke, myocardial infarction, coronary artery disease, peripheral arterial disease, systemic embolism, dementia, systolic/diastolic heart failure); social and family history (tobacco, alcohol, family history of stroke or AF); stroke severity/imaging (NIHSS on admission, ipsilateral ICA stenosis > 50%, intracranial arterial disease, lacunar pattern, stroke laterality); echocardiography (ejection fraction %, left atrial size, left atrial enlargement, cardiac shunt/PFO); risk score (CHA_2_DS_2_-VASc); and renal function and labs (eGFR, hemoglobin, platelet count, ALT, AST, HDL, LDL, HbA1c). Following manual chart review, no missing values were present for demographics and laboratory measures. For binary comorbidity indicators, records with unpopulated fields were adjudicated from notes/problem lists; when documentation did not support the condition, the indicator was coded absent. Non-numeric features were excluded, and numeric features were normalized to a range of 0–1. From an initial 223 cases reviewed based on their EHR, we identified 197 with ECG listed for extraction. Ultimately, 189 of these ECGs were successfully extracted and matched to the reviewed EHR record, forming our final study cohort. The sample size was fixed by cohort availability, with 49 PAF events.

### 2.4. Deep Learning Model Architecture

The proposed model consisted of a hybrid architecture designed to integrate ECG and EHR data for binary classification of PAF risk. For ECG feature extraction, a series of CNN layers processed the ECG signals to extract temporal features. Multi-head attention layers captured long-range dependencies across leads. The compressed ECG features were reduced to a fixed dimensionality (n_ECG_) using dense layers. For EHR feature processing, EHR features were processed through dense layers to reduce their dimensionality (n_EHR_) while preserving critical information. To clearly define the optimal balance between modalities, we systematically adjusted the representation (compression dimensions) of ECG and EHR inputs to identify their relative contributions to predictive performance.

Compressed ECG and EHR features were concatenated and passed through additional dense layers for final classification. The model architecture was optimized through hyperparameter tuning, including the number of attention heads (4–8), compression dimensions (n_ECG_ and n_EHR_), learning rate (10^−4^ to 10^−3^), and augmentation strategies. The tuning process specifically aimed to identify the optimal balance between ECG and EHR inputs for the best performance. Models were implemented in TensorFlow [16].

### 2.5. Training and Validation

The dataset was split into training and testing sets using 5-fold cross-validation to ensure robustness. Each experiment was repeated 10 times with different random seeds to assess variability in performance metrics. Data augmentation was performed during training with an augmentation probability (P_aug_ = 0.1) optimized for generalization without overfitting. The model was trained on NVIDIA RTX 6000 Ada GPU (Manufactured by NVIDIA Corporation, Santa Clara, CA, United States), using the Adam optimizer with an exponential learning rate decay schedule. Binary cross-entropy loss was used as the objective function. Test–time augmentation further improved prediction robustness by averaging predictions across augmented test samples. The training time was 50 GPU-hours. Decision curve analysis was not performed in this pilot due to limited events; clinical utility was not assessed.

### 2.6. Statistical Analysis

Model performance was evaluated using several metrics: accuracy, area under the receiver operating characteristic curve (AUROC), sensitivity, specificity, precision, and F1 score. The statistical significance of performance differences between models was assessed using paired *t*-tests. We also conducted pairwise Wilcoxon tests versus the best overall EHR contribution for the top performing repeats across all metrics. The significance threshold was considered *p* < 0.05. Feature importance analysis was conducted using Random Forest models to identify the most predictive variables in the EHR dataset. Additionally, the contribution of ECG versus EHR data was analyzed by varying their respective compression dimensions in the model and evaluating performance changes.

The study pipeline, as shown in Figure 1, consists of data preparation, an example of a preprocessed 12-lead ECG waveform, and multimodal data processing in the deep learning pipeline for PAF prediction.

## 3. Results

The study included 189 cryptogenic stroke patients, of whom 49 (26%) had a diagnosis of PAF. The mean age of the cohort was 71.4 years. Patients with PAF were significantly older than those without (75.4 years vs. 70.0 years, *p* = 0.004). The cohort was predominantly female (57.7%) and White (82.5%), with no significant differences in sex or race between the PAF and non-AF groups. The average monitoring duration for the PAF group was longer than for the non-AF group (22.6 months vs. 18.3 months), though this difference was not statistically significant (*p* = 0.064) (Table 1, Figure A1).

The results of this study demonstrate the effectiveness of a multimodal deep learning model that integrates ECG (denoised + band-pass filtered 0.5–40 Hz) and EHR data for predicting PAF. Using 189 stroke patients, the model was evaluated across multiple configurations and metrics, with a primary focus on the accuracy, sensitivity, and specificity. Each best performing configuration achieved an accuracy of 0.70 (SD: 0.04), sensitivity of 0.72 (SD: 0.42), and specificity of 0.87 (SD: 0.06) (Appendix A, Table A2). The large SD for some metrics reflects fold-to-fold variability driven by the small number of PAF events per fold.

This model compressed ECG to 32 and EHR to 16 latent dimensions and incorporated 8 attention heads into the Transformer architecture. Data augmentation with a probability of 0.1 further enhanced model generalization without overfitting. Figure 2 illustrates the overall distribution of performance metrics of different architectures and compares key performance metrics for the best model configuration.

Models using only EHR data achieved an accuracy of 0.67 (SD: 0.2; *p* < 0.05), a sensitivity of 0.72 (SD: 0.42; *p* < 0.05), and a specificity of 0.80 (SD: 0.32; *p* < 0.05), while those using only ECG data showed an accuracy of 0.52 (SD: 0.06; *p* < 0.05), a sensitivity of 0.51 (SD: 0.23; *p* < 0.05), and a specificity of 0.84 (SD: 0.07; *p* < 0.05). The integration of both data modalities not only enhanced overall predictive performance but also highlighted the critical role of achieving the right balance between ECG and EHR inputs. Systematic analysis demonstrated that predictive performance peaked when EHR data comprised approximately one-third of the model input (Figure 3, Table A3), underscoring the importance of balancing clinical data with electrophysiological information. Exploratory Wilcoxon tests versus the 33% EHR condition showed significant differences for nearly all comparisons (Appendix E, Table A4); the only non-significant pair was accuracy versus 67% (*p* = 0.33).

Figure 4 ranks EHR feature importance. Higher ranks included age, hemoglobin, and left-atrial measures, consistent with known PAF risk correlates.

## 4. Discussion

The findings from this study highlight the potential of multimodal deep learning models in predicting PAF by leveraging complementary information from ECG waveforms and EHR data. Although the dataset is not large and imbalanced, we employed 5-fold cross-validation and repeated the experiments 10 times to avoid potential overfitting. A novel aspect of this work is the direct comparison of ECG and EHR contributions, which reveals that an optimal balance is achieved when EHR data accounts for approximately 33% of the overall input. The observed improvement in predictive performance when combining these modalities aligns with prior studies [7,17,18]. For instance, Tang et al. demonstrated that integrating ECG with clinical features improved PAF recurrence prediction after catheter ablation [7]. Also, Khurshid et al. reported that combining ECG with clinical risk factors yielded complementary benefits for PAF prediction [19]. Recent studies suggest links between brain tissue susceptibility and vascular/arrhythmic risk [20,21]. In clinics, multimodal risk scores could be used to prioritize patients for extended ambulatory ECG and earlier cardiology follow-up.

Although our model’s accuracy is comparable to other studies, our critical insight is that optimizing the EHR data contribution significantly enhances model specificity and maintains clinical interpretability, a vital step for real-world applicability [6,22]. However, when considering the optimal contribution balance, the 33% input from EHR data emerges as a factor in attaining this performance level. This discrepancy may be attributed to differences in dataset size, patient demographics, or the transient nature of PAF, which present unique challenges for prediction.

The relatively low sensitivity observed in our study reflects the difficulty in detecting sporadic PAF episodes from limited data points, a limitation also noted in other works focusing on PAF [23]. Nonetheless, the high specificity indicates that our model is effective at ruling out low-risk cases, and our analysis suggests that the EHR contribution plays a pivotal role in enhancing specificity while maintaining an acceptable trade-off with sensitivity.

Regarding feature importance analysis (Figure 4 and Figure A2), hemoglobin levels emerged as an important feature, potentially reflecting underlying anemia, polycythemia [24], or other systemic conditions associated with PAF risk [18,25]. This further reinforces the value of the EHR component in our multimodal approach, underscoring that even a one-third contribution from EHR data can capture essential clinical nuances that improve overall model performance.

While our results are promising, several limitations warrant consideration. The primary limitation is the sample size of 189 patients from a single center, which may affect the generalizability of our findings. We observed instability in some metrics, which is attributable to the small pilot cohort and rare positives per CV fold. Accordingly, the retrospective nature of this pilot study requires our model to be validated in larger, more diverse populations, preferably through prospective studies. Given the pilot scope, we did not include full classical ML benchmarking; this is planned for a larger, prospective, multi-site validation. Furthermore, the relatively low sensitivity observed in our models reflects the inherent difficulty of predicting sporadic PAF episodes from limited data. Future work should focus on external validation to confirm our findings and refine the optimal 33% EHR data contribution as a benchmark for improving predictive accuracy and sensitivity (e.g., class-weighted or focal loss) in different clinical settings. Also, medication exposures (e.g., aspirin, statin) were not used at index to avoid post-stroke initiation bias; they are planned for future prospective cohorts.

## 5. Conclusions

This pilot study provides evidence supporting the use of multimodal deep learning for predicting paroxysmal atrial fibrillation among stroke patients by combining ECG and EHR data. Our analysis uniquely reveals that optimal performance is achieved when EHR data contributes approximately 33% of the overall input, underscoring the critical importance of balancing heterogeneous data sources. By leveraging complementary information from these modalities, our approach offers a scalable solution for early risk stratification and intervention in clinical practice. Future research should validate our findings externally, dynamically optimize the balance between EHR and ECG data, and explore real-time clinical deployment to enhance early PAF detection, clinical decision making, and patient outcomes.

## Figures and Tables

**Figure 1 bioengineering-12-00961-f001:**
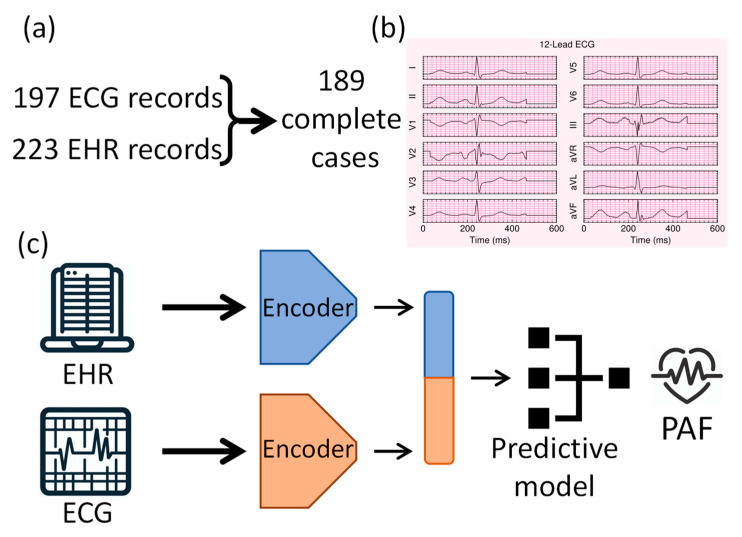
Overview of the study pipeline, including (**a**) data preparation steps with patient inclusion numbers, (**b**) an example of a preprocessed 12-lead ECG waveform, and (**c**) the deep learning pipeline integrating multimodal ECG and EHR data for paroxysmal atrial fibrillation (PAF).

**Figure 2 bioengineering-12-00961-f002:**
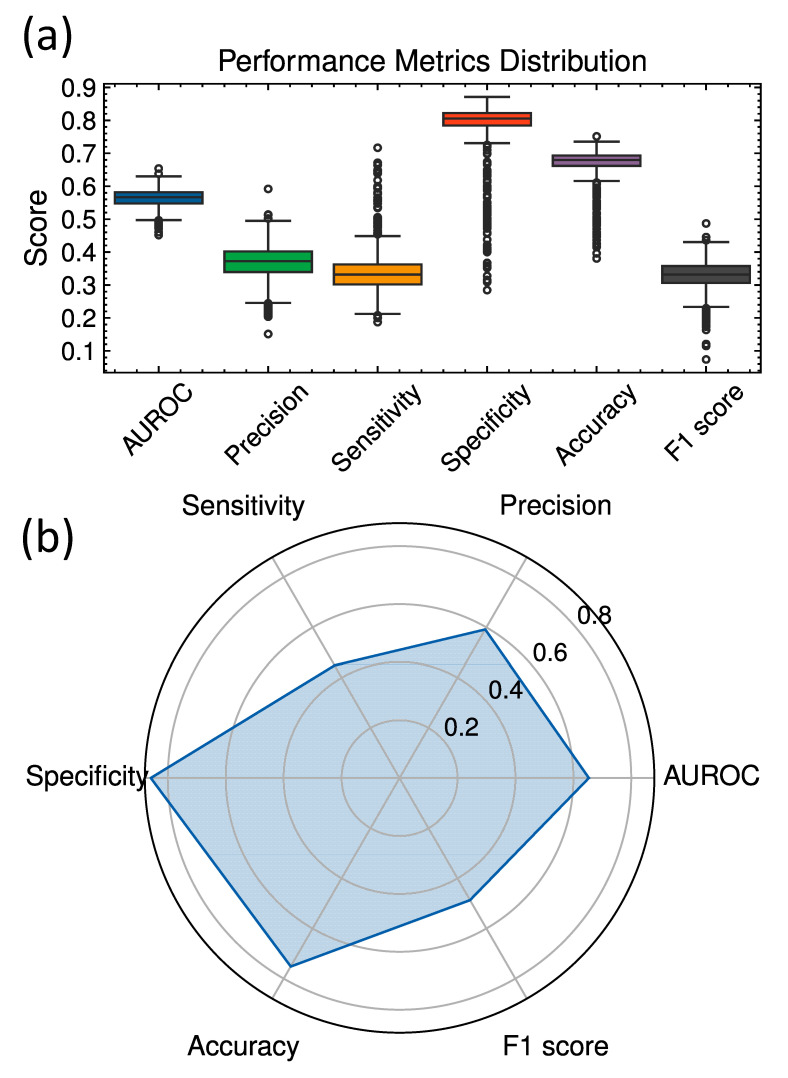
(**a**) Performance metrics across different deep learning architectures. (**b**) Spider plot illustrating the performance metrics (AUROC, sensitivity, specificity, precision, accuracy, and F1 score) for the best performing model configuration. The plot highlights the balanced trade-offs achieved across all six metrics.

**Figure 3 bioengineering-12-00961-f003:**
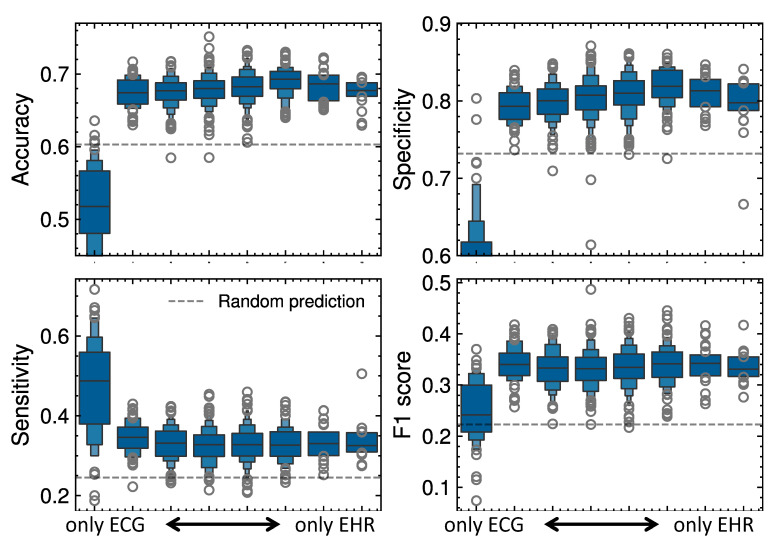
Impact of varying the relative contribution of EHR versus ECG data on predictive performance metrics. Moving left along the x-axis increases ECG data’s contribution (reducing EHR), whereas moving right increases EHR data’s contribution. For comparison, the dashed line represents baseline performance (dummy classifier).

**Figure 4 bioengineering-12-00961-f004:**
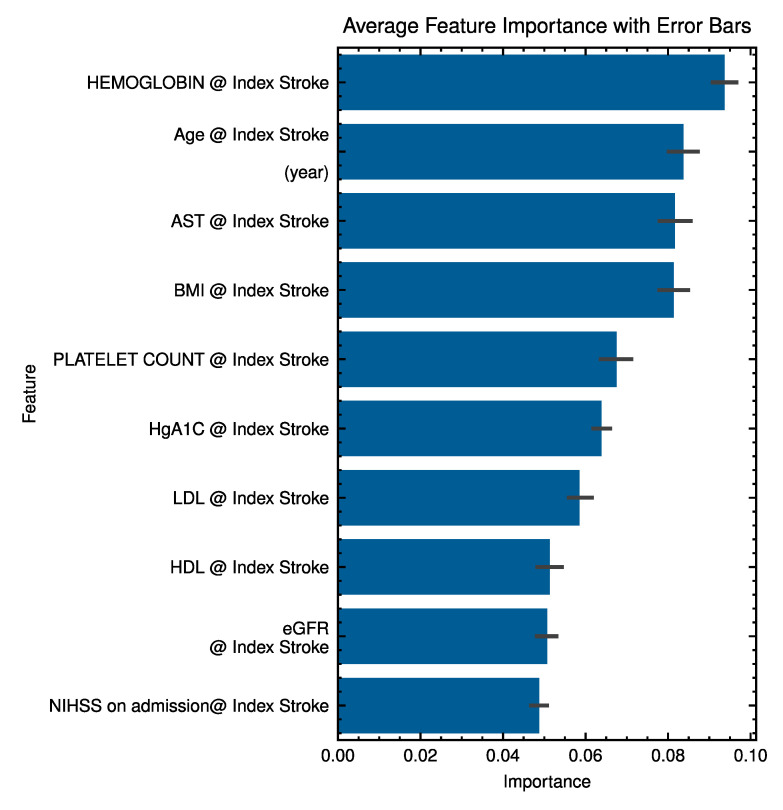
Feature importance analysis identifying key predictors in the EHR dataset. The error bars represent variability through training random seeds. The symbol ‘@’ indicates measurements obtained at the index stroke.

**Table 1 bioengineering-12-00961-t001:** Baseline demographic and clinical characteristics of the study population, stratified by paroxysmal atrial fibrillation (PAF) diagnosis. The *p*-values show the significant difference between PAF and no PAF.

Variable	Total (*n* = 189)	No PAF (*n* = 140)	PAF (*n* = 49)	*p*-Value
Age (years)	71.4 ± 11.4	70.0 ± 11.6	75.4 ± 9.6	0.004
Sex, ***n*** (%)				0.452
Male	80 (42.3)	62 (44.3)	18 (36.7)	
Female	109 (57.7)	78 (55.7)	31 (63.3)	
Race, ***n*** (%)				0.205
White	156 (82.5)	116 (82.9)	40 (81.6)	
Black	17 (9.0)	13 (9.3)	4 (8.2)	
Asian	1 (0.5)	0 (0.0)	1 (2.0)	
Others	14 (7.4)	11 (7.9)	3 (6.1)	
Ethnicity, ***n*** (%)				0.999
Hispanic	3 (1.6)	2 (1.4)	1 (2.0)	
Non-Hispanic	186 (98.4)	138 (98.6)	48 (98.0)	
Monitoring (months)	19.4 ± 13.9	18.3 ± 13.6	22.6 ± 14.5	0.064

## Data Availability

The data that support the findings of this study are not publicly available due to institutional policies and to protect patient privacy and confidentiality.

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
