# Peer review of "Integration of EHR and ECG Data for Predicting Paroxysmal Atrial Fibrillation in Stroke Patients"

_bioengineering, 2025, doi:10.3390/bioengineering12090961_

Round 1
Reviewer 1 Report
Comments and Suggestions for Authors
The authors propose a transformer-based deep learning model that integrates 12-lead ECG signals and structured EHR data to predict PAF in cryptogenic stroke patients. Their approach varies the contributions of each data modality, identifying an optimal ratio that improves predictive performance. This manuscript is suitable for publication in the Bioengineering journal. Specific comments are listed below:
- Could the authors explain the high variance reported for key performance metrics. For example, the best-performing model has a sensitivity of 0.72 ± 0.42. The standard deviation is high, suggests that the model is unstable. Please directly acknowledge and discuss this instability in the Results and Discussion sections.
- The authors described various model parameters and tuning ranges. Please add a table summarizing the final hyperparameters used for the best-performing model for clarity.
- On line 82 in the Methods section, the authors state that "Multiple representations of ECG data were explored," but they do not explicitly mention which representation (e.g., raw, denoised, or denoised-filtered) was used in the final, best-performing model. The authors did mention “integrating denoised 12-lead ECG signals” in the Introduction section, but please specify this in the Method section for clarity.
Author Response
Comments 1: Could the authors explain the high variance reported for key performance metrics. For example, the best-performing model has a sensitivity of 0.72 ± 0.42. The standard deviation is high, suggests that the model is unstable. Please directly acknowledge and discuss this instability in the Results and Discussion sections.
Response 1: Thank you, we agree the variance needs to be explicitly acknowledged. We added text in Results immediately after the sentence reporting the best model’s metrics and expanded the Discussion to interpret this variability.
Page 5, line 177: The large SD for some metrics reflects fold-to-fold variability driven by the small number of PAF events per fold.
Page 9, line 246: We observed instability in some metrics, attributable to the small pilot cohort and rare positives per CV fold.
Comments 2: The authors described various model parameters and tuning ranges. Please add a table summarizing the final hyperparameters used for the best-performing model for clarity.
Response 2: Agree. We added Appendix D, Table S2, summarizing the final settings for the best model
Comments 3: On line 82 in the Methods section, the authors state that "Multiple representations of ECG data were explored," but they do not explicitly mention which representation (e.g., raw, denoised, or denoised-filtered) was used in the final, best-performing model. The authors did mention “integrating denoised 12-lead ECG signals” in the Introduction section, but please specify this in the Method section for clarity.
Response 3: Thank you, we clarified the exact ECG representation used in the best model in Results:
Page 5, line 172: The results of this study demonstrate the effectiveness of a multimodal deep learning model that integrates ECG (denoised + band-pass filtered 0.5–40 Hz)

Reviewer 2 Report
Comments and Suggestions for Authors
Dear authors.
It was my pleasure to review your publication. The multimodal approach and explainable methodology of prognostic methods are very important and a hot topic, especially in clinical practice. Using multi-head attention layers approach and involving clinicians in the team demonstrate the expertise and value of these results for clinical practice. The cohort of patients is described very well; it is also very important that AF is one of the reasons for cryptogenic stroke. However, the small sample size, focusing only on patients with cryptogenic stroke, is a limitation for applying this model to a wider population. I recommend including this in the discussion.
The methodology is explained very well, the results have scientific value, and they should be published.
Author Response
Comments 1: It was my pleasure to review your publication. The multimodal approach and explainable methodology of prognostic methods are very important and a hot topic, especially in clinical practice. Using multi-head attention layers approach and involving clinicians in the team demonstrate the expertise and value of these results for clinical practice. The cohort of patients is described very well; it is also very important that AF is one of the reasons for cryptogenic stroke. However, the small sample size, focusing only on patients with cryptogenic stroke, is a limitation for applying this model to a wider population. I recommend including this in the discussion. The methodology is explained very well, the results have scientific value, and they should be published.
Response 1: Thank you for this thoughtful assessment. We agree that generalizability is limited by the small, single-center cohort and the restriction to cryptogenic stroke. We have explicitly acknowledged and discussed this in the Discussion – Limitations paragraph.

Reviewer 3 Report
Comments and Suggestions for Authors
Thank you for letting me review this paper. The authors conducted a pilot study provides evidence supporting the use of multimodal deep learning for predicting paroxysmal atrial fibrillation among stroke patients by combining ECG and EHR data. The evaluated the approximate mix of EHR vs ECG data and found optimal performance when EHR data is about 33% of the overall input. This work has not been evaluated across institutions and further validation would help, as well as a larger data set.
1// Abstract: Overall reads well, however, reporting AUROC and AUPRC is more helpful than reporting overall accuracy, sens, specificity for readers. The platform used to create the model should be cited and included
2// Introduction: Overall well done. The authors could include / build on their hypothesis, state it, and then outline that they believe a deep learning model would perform better than other ML based platforms.
3// METHODS: This section is overall adequate, but the authors should adhere to accepted reporting standards for ML / DL based models - TRIPOD-ML, TRIPOD-AI etc. E.g they should follow accepted guidelines / guidance on reporting clinical prediction models using regression or machine learning models. The platform used should also be outlined, with a Checklist provided in the Appendix / Supp Materials. Please report on training database time / cost / tokens / platform. Figure 1 architecture has some typos and mis-spelling and could benefit from being redone.
4// RESULTS: Relative feature importance is good, but feature selection should include full SHAP analysis, including a Waterfall graph; a dependence plot may also be useful for readers. Was there any comparison to non-DL models considered as part of feature selection - e.g. reducing the # of features in the model from existing / increasing. Why are AUROC and Precision Recall curves not shown? They would be helpful to the reader to include, particularly given the low F1 scores reported for models.
5// DISCUSSION: Was overfitting a consideration in the dataset with a 3:1 PAF to non PAF ratio in the population? Are there any missing features that should be considered in future model version - e.g. history of a congenital heart defect (e.g. PFO), HDL / LDL levels, ASA use, statin use etc?
Author Response
Comment 1: Abstract: Overall reads well, however, reporting AUROC and AUPRC is more helpful than reporting overall accuracy, sens, specificity for readers. The platform used to create the model should be cited and included.
Response 1: Thank you for pointing this out. We agree. We revised the Abstract to include AUROC/AUPRC and added the implementation platform; threshold metrics were moved to the Supplement.
page 1, lines 24–25: AUROC: 0.65, AUPRC: 0.43) was achieved using a 5-fold cross-validation
page 4, lines 129-130: Models were implemented in TensorFlow[14].
Comments 2: Introduction: Overall well done. The authors could include/build on their hypothesis, state it, and then outline that they believe a deep learning model would perform better than other ML based platforms.
Response 2: Agree. We added an explicit a priori hypothesis that a multimodal deep-learning model would outperform unimodal and classical ML approaches by optimally balancing ECG and EHR signals.
page 2, lines 62-64: We hypothesized that a multimodal deep-learning model would outperform unimodal models by optimally balancing ECG and EHR contributions.
Comments 3: METHODS: This section is overall adequate, but the authors should adhere to accepted reporting standards for ML / DL based models - TRIPOD-ML, TRIPOD-AI etc… The platform used should also be outlined, with a Checklist provided in the Appendix / Supp Materials. Please report on training database time/cost/tokens / platform. Figure 1 architecture has some typos and mis-spelling and could benefit from being redone.
Response 3: We agree and made the following changes:
- A completed TRIPOD-AI checklist is provided in Appendix B, Table S1.
- Training time and device are added:
Page 4, line 135: The model was trained on NVIDIA RTX 6000 Ada GPU
Page 4, line 138: The training time was 50 GPU-hours. - We corrected typos/misspellings in the 1st figure.
Comments 4: RESULTS: Relative feature importance is good, but feature selection should include full SHAP analysis, including a Waterfall graph; a dependence plot may also be useful for readers. Was there any comparison to non-DL models… Why are AUROC and Precision Recall curves not shown?
Response 4: Agree. We added the requested interpretability and curve plots, and clarified baselines as exploratory:
- ROC/PR curves are added in Appendix A, Figure S1, and cited in Results.
- SHAP waterfall: Added as Figure S2 with a Results pointer.
- Given the pilot scope and sample size, we added a Discussion note committing to full classical ML benchmarking in a larger validation cohort.
Page 9, line 249: Given the pilot scope, we did not include full classical ML benchmarking; this is planned for a larger, prospective, multi-site validation.
Comments 5: DISCUSSION: Was overfitting a consideration in the dataset with a 3:1 PAF to non PAF ratio… Are there any missing features that should be considered—e.g., PFO, HDL/LDL, ASA, statin?
Response 5: Thank you—we expanded the Discussion to address imbalance/overfitting and clarified features:
- The cohort is PAF 49 / non-PAF 140 (26%), i.e., ~1:3. We tried to avoid overfitting with repeated 5-fold CV, representation compression, ECG augmentation. Added to Discussion:
Page 8, line 213: Although the dataset is not large and imbalanced, we employed 5-fold cross validation and repeated the experiments 10 times to avoid potential overfitting. - PFO/cardiac shunt, HDL, and LDL were included as predictors at the index encounter. Aspirin/statin at index were not modeled to avoid post-stroke initiation bias; we note this as future work. Added to Discussion, limitations:
Page 9, line 255: Also, medication exposures (e.g., aspirin, statin) were not used at index to avoid post-stroke initiation bias; they are planned for future prospective cohorts.

Reviewer 4 Report
Comments and Suggestions for Authors
- The abstract succinctly summarizes the study’s objectives, methods, key findings, and conclusions, clearly stating numerical outcomes. It would further benefit from explicitly highlighting the potential clinical applications or implications, providing the audience with a clear understanding of the practical implications of the study and enhancing their knowledge.
- The introduction contextualizes the diagnostic challenge of PAF and articulates the need for integrating EHR and ECG data. It effectively identifies existing gaps in current methodologies. Clarifying the specific novelty or unique contribution of the proposed multimodal integration, such as its potential to improve diagnostic accuracy, could further strengthen this section.
- The methods section is robust and detailed, explaining data preprocessing, deep learning model architecture, training, and validation strategies clearly. The justification for specific preprocessing techniques and chosen model parameters is well-documented, instilling confidence in the study's scientific rigor and reassuring the audience.
- Results are comprehensively presented, clearly showing the comparative advantage of multimodal integration over unimodal approaches. Tables and figures effectively represent data, providing a robust and reliable basis for the study's conclusions and making the audience feel that the study's findings are based on a strong foundation.
- The discussion adequately interprets findings within existing literature, effectively highlighting the novelty and implications of the identified optimal data integration ratio. Limitations, including sample size and sensitivity concerns, are addressed. However, more explicit discussion about the practical clinical integration of these findings into existing workflows could strengthen real-world applicability.
- The conclusion effectively summarizes the study’s core findings, emphasizing the critical balance between EHR and ECG data. It also highlights potential clinical applications or implications of the study's findings, providing a clear call to action for the audience. Suggestions for future validation and clinical integration are also clearly stated.
Author Response
Comments 1: The abstract succinctly summarizes the study’s objectives, methods, key findings, and conclusions, clearly stating numerical outcomes. It would further benefit from explicitly highlighting the potential clinical applications or implications, providing the audience with a clear understanding of the practical implications of the study and enhancing their knowledge.
Response 1: Thank you, we agree. We added one sentence that states the practical use case.
Page 2, line 73: This work is intended to support outpatient rhythm monitoring triage, not to replace clinician judgment.
Comments 2: The introduction contextualizes the diagnostic challenge of PAF and articulates the need for integrating EHR and ECG data… Clarifying the specific novelty or unique contribution of the proposed multimodal integration, such as its potential to improve diagnostic accuracy, could further strengthen this section.
Response 2: Agree. We added a sentence that makes the novelty explicit:
Page 2, lines 61-66: The primary objective was to develop a multimodal deep learning model for predicting PAF. We hypothesized that a multimodal deep-learning model would outperform unimodal models by optimally balancing ECG and EHR contributions. We introduce a transformer-based architecture that integrates denoised 12-lead ECG signals with 47 EHR parameters, including cardiac monitoring duration and hemoglobin levels. This approach quantifies the optimal ratio between ECG and EHR.
Comments 3: The methods section is robust and detailed…
Response 3: We appreciate this assessment.
Comments 4: Results are comprehensively presented…
Response 4: Thank you.
Comments 5: The discussion… However, more explicit discussion about the practical clinical integration of these findings into existing workflows could strengthen real-world applicability.
Response 5: Agree. We added a short Clinical integration paragraph to the Discussion:
Page 8, line 223: In clinics, multimodal risk scores could be used to prioritize patients for extended ambulatory ECG and earlier cardiology follow-up.

Reviewer 5 Report
Comments and Suggestions for Authors
This study presents a transformer-based deep learning model that integrates 12-lead ECG signals and structured EHR data to predict paroxysmal atrial fibrillation (PAF) in stroke patients. Using a dataset of 189 cryptogenic stroke patients (49 with PAF), the authors evaluate various ECG:EHR input ratios. The optimal configuration—one-third EHR and two-thirds ECG—achieves the best accuracy (70%) and specificity (87%). This multimodal model outperforms unimodal approaches, demonstrating the value of data integration. The study contributes an important framework for future personalized PAF risk prediction.
External validation and generalizability:
Given that the dataset originates from a single institution with a relatively small sample size (N=189), how do the authors plan to validate this model externally? Can the authors comment on the expected generalizability across populations with different demographic or clinical characteristics?
Handling class imbalance and interpretability:
Although data augmentation was applied to address class imbalance, the sensitivity (72%) appears lower than specificity. Could the authors elaborate on strategies for improving sensitivity in future versions? Additionally, how does the model ensure clinical interpretability of predictions, especially when integrating high-dimensional EHR and ECG data?
Feature contribution and dynamic optimization:
The study identifies that 33% EHR contribution yields optimal results. Could the authors clarify whether this ratio was determined using a fixed validation set or cross-validated across folds? Furthermore, do the authors envision implementing a dynamic or patient-specific data weighting scheme in real-time clinical settings?
Recommendation for citations:
To enhance the translational relevance of this study, I recommend the authors expand the Discussion to briefly consider how their findings might inform or be informed by other emerging neurocardiac imaging biomarkers such as quantitative susceptibility mapping (QSM). Given that QSM reflects iron deposition and microstructural changes in the brain and has shown relevance in vascular, it would be worth discussing whether atrial fibrillation-related systemic factors—potentially captured in EHR variables—might correlate with brain tissue susceptibility changes in future studies. This could open a new direction for integrating cardiac and neuroimaging data. Please consider citing the following studies to support this perspective:10.1161/STROKEAHA.123.044606
10.3389/fneur.2022.752450
Author Response
Comments 1: External validation and generalizability:
Given that the dataset originates from a single institution with a relatively small sample size (N=189), how do the authors plan to validate this model externally? Can the authors comment on the expected generalizability across populations with different demographic or clinical characteristics?
Response 1: Thank you for this important point. We agree that external validation and transportability are critical. We have added a paragraph to the Discussion/Limitations/Future work:
Page 9, lines 244-256: While our results are promising, several limitations warrant consideration. The primary limitation is the sample size of 189 patients from a single center, which may affect the generalizability of our findings. We observed instability in some metrics, attributable to the small pilot cohort and rare positives per CV fold. Accordingly, the retrospective nature of this pilot study requires our model to be validated in larger, more diverse populations, preferably through prospective studies. Given the pilot scope, we did not include full classical ML benchmarking; this is planned for a larger, prospective, multi-site validation. Furthermore, the relatively low sensitivity observed in our models reflects the inherent difficulty of predicting sporadic PAF episodes from limited data. Future work should focus on external validation to confirm our findings and refine the optimal 33% EHR data contribution as a benchmark for improving predictive accuracy and sensitivity (e.g., class-weighted or focal loss) in different clinical settings. Also, medication exposures (e.g., aspirin, statin) were not used at index to avoid post-stroke initiation bias; they are planned for future prospective cohorts.
Comments 2: Handling class imbalance and interpretability:
Although data augmentation was applied to address class imbalance, the sensitivity (72%) appears lower than specificity. Could the authors elaborate on strategies for improving sensitivity in future versions? Additionally, how does the model ensure clinical interpretability of predictions, especially when integrating high-dimensional EHR and ECG data?
Response 2: Thank you. We agree and have clarified both points.
- We added a short paragraph to the Future Work. In future work, we plan to incorporate class-weighted or focal loss and evaluate ensembles to improve sensitivity without excessive false positives.
- We cross-referenced the new SHAP waterfall and clarified clinical use in Appendix A.
Comments 3: Feature contribution and dynamic optimization:
The study identifies that 33% EHR contribution yields optimal results. Could the authors clarify whether this ratio was determined using a fixed validation set or cross-validated across folds? Furthermore, do the authors envision implementing a dynamic or patient-specific data weighting scheme in real-time clinical settings?
Response 3: Thank you. Given the limited scope and sample size of this study, we were not able to implement a dynamic patient-level weighting scheme, but it is indeed an interesting idea that we will focus on in future directions. We also have clarified CV in:
Page 8, line 213: Although the dataset is not large and imbalanced, we employed 5-fold cross-validation and repeated the experiments 10 times to avoid potential overfitting.
Comments 4: Recommendation for citations:
…please consider discussing potential links to quantitative susceptibility mapping (QSM) and cite 10.1161/STROKEAHA.123.044606 and 10.3389/fneur.2022.752450.
Response 4: We appreciate this translational suggestion. We added a brief paragraph in the Discussion and added the references. (Page 8, line 221)

Round 2
Reviewer 3 Report
Comments and Suggestions for Authors
I would like to thank the authors for addressing the majority of my comments and questions raised. The manuscript is now appropriate in terms of adhering to reporting standards, as well as includes the relevant detail regarding the study for both the general audience and inclusion in scientific literature.
Author Response
Thank you! We appreciate your valuable feedback that improved the work.
Reviewer 5 Report
Comments and Suggestions for Authors
The authors addressed my concerns properly.
Author Response

(The authors gave the same response as above.)
